# Raman Study of 532-Nanometer Laser-Induced Degradation of Red Lead

**DOI:** 10.3390/ma17040770

**Published:** 2024-02-06

**Authors:** Yan Li, Junjie Ma, Kang He, Fengping Wang

**Affiliations:** Department of Physics, School of Mathematics and Physics, University of Science and Technology Beijing, Beijing 100083, China; liyan000998@163.com (Y.L.); b20150328@xs.ustb.edu.cn (K.H.)

**Keywords:** red lead, confocal micro-Raman, degradation laser power, photothermal effect

## Abstract

Red lead is commonly employed as a red pigment in numerous valuable cultural artifacts. Raman spectrometry has been widely employed as the primary tool in many nondestructive studies on red lead. Therefore, it is necessary to evaluate and study the impact of lasers on the pigment. The degradation of red lead induced by a 532 nm laser is investigated using micro-Raman spectroscopy. At room temperature, red lead begins to degrade into β-PbO when the power density of the 532 nm laser reaches approximately 5.1 × 10^4^ W/cm^2^ (laser: 532 nm, objective: 50×). At this point, the temperature at the focus of the sample is estimated to be at least 500 °C, aided by the Raman peak shift of β-PbO. Furthermore, the power density of the laser-induced degradation decreases as the temperature of the red lead increases. Hence, the degradation of red lead can be attributed to the photothermal effect. The temperature rise can be explained by two factors. First, red lead exhibits a high absorbance of approximately 0.5942 at 532 nm. Second, red lead has significantly low thermal diffusivity and conductivity, measuring 0.039 mm^2^·s^−1^ and 0.078 W·m^−1^·K^−1^, respectively, which leads to heat accumulation at the focal point of the laser beam. To better preserve cultural heritage, the appropriate laser power should be prioritized when the degradation process is caused by the thermal effect of laser irradiation.

## 1. Introduction

Currently, nondestructive methods such as Raman spectroscopy, reflectance spectroscopy, and X-ray fluorescence spectroscopy are widely used in the field of cultural heritage for material characterization, conservation state assessment, understanding artist’s techniques, and determining the most suitable conservation plan. Raman spectroscopy is an important technique for material characterization within the realm of cultural heritage conservation. However, the employment of lasers poses a potential risk of damaging the artifacts. Therefore, it is crucial to assess and prevent laser-induced damage to cultural relics.

Red lead (Pb_3_O_4_) is a mixed-valence compound consisting of lead atoms coordinated to oxygen atoms in both octahedral and pyramidal configurations, resulting in valences IV and II [1]. Red lead, also known as minium, is one of the earliest artificially prepared pigments, produced by calcination of lead white (hydrocerussite, 2Pb(CO_3_)·Pb(OH)_2_). Red lead has been commonly found in various artworks, including wall paintings, colored drawings, and oil paintings [2,3,4]. Despite its widespread use, red lead has degraded over time into black materials due to natural and human factors. Many studies have been conducted to investigate the degradation of red lead, with the black material initially thought to be plattnerite [5,6]. However, Raman analysis of red lead using 514 and 466 nm lasers revealed the presence of Raman bands of β-PbO, leading Burgio to conclude that β-PbO is the degradation product of red lead, ascribed to the photothermic effect [7]. However, β-PbO was not detected when using the 785 and 1064 nm lasers as excitation sources during the Raman analysis of red read. These findings suggest that the thermal effect of the laser on red lead was dependent on the wavelength of the laser. Costantini investigated the degradation of plattnerite under a 633 nm laser and observed the formation of red lead and β-PbO when the laser power was increased [8]. They hypothesized that the irradiation time was negligible, and within a few seconds, the local temperature of the irradiated material rapidly increased until it reached a maximum corresponding to a stable thermal regime.

Red lead is commonly reduced to β-PbO at temperatures over 500 °C [9]. However, it is uncertain whether the local temperature of red lead reaches 500 °C when using relatively low laser power. Usually, the laser energy required for photothermal degradation is relatively high. This is because the generation of photothermal effects requires sufficient energy to excite the vibrational modes in the material molecules and convert them into heat energy, thus raising the temperature inside the sample [10]. Therefore, measuring the in situ temperature at the focus of the laser is crucial in confirming the role of the laser in the degradation process of red lead. Traditional methods such as thermocouples and infrared thermometers are inadequate for this purpose due to the small diameter of the laser focus (approximately 1.5 μm) and the fact that the degradation area exists only on the surface of red lead. Therefore, a new method must be employed to measure the in situ temperature. Recently, Raman scattering has become a suitable method for determining the temperature at the focus of a laser beam on samples.

To better understand the mechanism of red lead degradation caused by lasers, and to protect cultural relics more effectively, we determined the laser power density causing red lead degradation (532 nm) and the temperature required for thermal degradation (785 nm) using micro-Raman spectroscopy. Simultaneously, the temperature at the focal area during laser-induced red lead degradation was characterized, confirming that the degradation caused by the laser is attributable to the photothermal effect. By measuring the absorption spectrum of red lead and its thermal diffusivity coefficient, the degradation mechanism of lead white caused by low-power lasers was well explained.

## 2. Materials and Methods

### 2.1. Materials

Red lead and β-PbO (95% pureness) were purchased from Chengdu Jinshan Chemical Reagent Co., Ltd. (Chengdu, China), and were analyzed in powder form by taking a small amount from the commercial container and placing it on a glass sample holder. The mass ratio of red lead to β-PbO was 10:1 in the mixed compounds used to collect Raman spectra under different laser power densities (532 nm). Pellets of red lead and β-PbO were prepared using a thermal cell (Linkam stage) and placed on a circular glass coverslip (13 × 0.1 mm), which was directly in contact with a heating plate that propagated heat to the samples.

### 2.2. Instruments

Raman spectra were collected using a HR800 spectrometer ( HORIBA Jobin Yvon, Paris, France) interfaced with an Olympus microscope equipped with two semiconductor lasers (532 nm and 785 nm) as the excitation source and a CCD detector cooled by a Peltier device. In this study, only the 50× objective was used to collect the spectra. Here, all the samples were placed on the focal plate of the laser beam. The heating process was carried out using a Linkam stage, which can operate over a temperature range of 25 to 1000 °C.

The diffuse reflection spectrum was collected from 400 to 800 nm using a Lamda 980 visible–ultraviolet spectrophotometer. The thermal diffusivities of red lead and β-PbO at room temperature were measured using a Netzsch laser thermal conductivity testing instrument (LFA467, Bavaria, Germany). The sample thickness was 3.3080 mm, and the laser voltage and pulse width were set to 260 V and 1.5 ms, respectively. To ensure the accuracy of the thermal diffusivities, six temperature points were measured.

### 2.3. The Design of the Experiment

The light irradiation system used in the laser degradation experiment was the incident light system of the Raman spectrometer (532 nm). The minimum theoretical spot diameter for the 532 nm excitation laser is about 1.5 μm when used with an Olympus 50× objective. The laser power can be adjusted continuously via an optical system composed of a half-wave plate, a Glan prism, a stepper motor rotation mount, and related control software. During the experiment, all the samples were placed on the focal plate of the laser beam and irradiated for 10 s. During the experiment, all the samples were placed on the focal plane of the laser beam and irradiated for 10 s. The power density (W/cm^2^) of the laser beams can be controlled continuously from 56 W/cm^2^ to 5.7 × 10^4^ W/cm^2^, as shown in Appendix A.

Firstly, standard spectra were collected for red lead and β-PbO under different exciting sources. Secondly, the Raman spectra of red lead under different laser power densities (532 nm) were acquired to explore the degradation process of red lead caused by laser irradiation. Thirdly, the Raman spectra of red lead were obtained under a laser power density of 2.3 × 10^4^ W/cm^2^ (532 nm) for a duration of 7200 s to determine if red lead is stable when the laser energy is lower than the degradation energy. Fourthly, the Raman spectra of red lead were collected from 30 °C to 650 °C under a 785 nm laser. Fifthly, the Raman spectra of red lead under different laser power densities (532 nm) at different temperature stages were collected to predict the degradation temperature of red lead. Sixthly, the Raman spectra of β-PbO were collected from 30 °C to 600 °C under a 532 nm laser to establish the relationship between Raman peak position and temperature. Seventhly, the Raman spectra of a mixture with a mass ratio of β-PbO and red lead at 1:10 were acquired under different laser power densities (532 nm) to determine the peak position of β-PbO when the laser power density reached the transformation power density. Eighthly, the absorption spectra and thermal diffusivities of red lead and β-PbO were measured to help explain the degradation mechanism. The related parameters of Raman measurement during the experiment are shown in Table 1.

## 3. Results

### 3.1. The Stokes Raman Spectra of Red Lead and β-PbO Excited by 532 nm and 785 nm Lasers

In this work, we adopted two excitation sources (532 nm and 785 nm) to carry out the Raman measurement. Therefore, the Raman spectra of red lead and β-PbO were obtained firstly under the two excitation wavelengths.

The Raman spectra of red lead and β-PbO under different excitation wavelengths (785 nm and 532 nm) are presented in Figure 1. When using the 532 nm laser as the excitation source, the Raman bands of red lead are located at 163, 478, and 547 cm^−1^ (Figure 1a). However, the Raman bands of red lead are located at 120, 150, 312, 389, and 548 cm^−1^ when using the 785 nm laser as the excitation source (Figure 1b). The Raman bands (142, and 289 cm^−1^) of β-PbO are nearly the same under the 532 nm and 785 nm excitation (Figure 1c,d).

### 3.2. The In Situ Temperature Estimated by the Raman Spectra under Different Laser Power Densities

The Raman spectra of red lead were found to vary with changes in laser power density at room temperature. Figure 2 shows the Raman spectra of red lead under various laser power densities (532 nm). From 3.2 × 10^4^ W/cm^2^ to 4.4 × 10^4^ W/cm^2^, the Raman spectra showed peaks at 162, 478, and 546 cm^−1^, which were attributed to red lead (Figure 2a–c). When the laser power density reached 5.1 × 10^4^ W/cm^2^, a new peak at 136 cm^−1^ was observed (Figure 2d), and at 5.4 × 10^4^ W/cm^2^, another new peak at 270 cm^−1^ was observed (Figure 2e). These two peaks may possibly belong to β-PbO at high temperatures. Under a lower laser power density (1.5 × 10^3^ W/cm^2^), these two new peaks moved to 142 cm^−1^ and 289 cm^−1^, respectively, which belongs to β-PbO (Figure 2f). These findings suggest that red lead transformed into β-PbO when the laser power density exceeded ~5.1 × 10^4^ W/cm^2^. In some research, red lead can be obtained by keeping β-PbO at 535 °C for 176 h [11]. However, the time is very short (~60 s) which is not sufficient for the oxidation of β-PbO. Therefore, the irreversible changes in the Raman spectra are nearly unobservable.

In order to determine whether red lead can stably exist under low-power laser excitation, the Raman spectra of red lead were collected at 100 s intervals under a 2.3 × 10^4^ W/cm^2^ laser power density for a total irradiation time of 7200 s. The related Raman mapping is shown in Figure 3. It is obvious that the peak positions and intensities remain stable during the irradiation time. This indicates that red lead is very stable under low power laser excitation, as long as the laser power is lower than the degradation threshold (5.1 × 10^4^ W/cm^2^). Photochemical reactions start from excited states and, distinct from the photophysical processes, convert the substrate molecule into product(s). Generally, the initial photochemical products are unstable and undergo secondary thermal and/or photochemical reactions, yielding more stable photoproduct(s). In photochemistry, reactions initiated from ground electronic states are thermal reactions, because their rates depend on temperature, distinct from photochemical reaction rates. Photothermal reactions lead to product generation only when thermodynamic and kinetic requirements are satisfied [12]. Here, the discovery of the degradation threshold appears to meet the thermodynamic and kinetic requirements. Therefore, it is very likely that the degradation process is a photothermal process rather than a photochemical one.

To determine the transformation temperature of red lead, in situ Raman measurements were conducted on red lead at temperatures ranging from 30 °C to 650 °C, with a 5 °C step interval. A 785 nm laser was utilized as the excitation source in order to eliminate the thermal effects caused by the 532 nm laser. The Raman spectra are shown in Figure 4. The Raman bands of red lead were clearly visible and shifted towards lower wavenumbers when the temperature was below 485 °C. At 495 °C, the Raman band at 138.1 cm^−1^ appeared, indicating that Pb_3_O_4_ began to degrade into β-PbO. As the temperature increased from 495 °C to 530 °C, the band at 138.1 cm^−1^ became stronger and shifted down to 136.6 cm^−1^, while the band at 113.7 cm^−1^ weakened. When the temperature was above 530 °C, only the band belonging to β-PbO was observed, indicating that red lead had completely transformed into β-PbO. At 30 °C, only the bands at 142 and 289 cm^−1^ were observed, further confirming that β-PbO is the degradation product of red lead. Therefore, based on the experimental results, it can be concluded that red lead begins to degrade at approximately 495 °C. Meanwhile, the degradation product is the same for the laser-induced transformation and thermal transformation of red lead.

If the degradation of red lead due to laser irradiation is caused by the photothermal effect, the temperature of the laser irradiation area should exceed 495 °C when the laser power density is 5.1 × 10^4^ W/cm^2^ (532 nm). Therefore, obtaining the in situ temperature of the laser focus area is critical. Two methods were employed to determine the in situ temperature. The first method involved predicting the transformation temperature of red lead by using the laser power densities required for transformation at different temperatures. The second method involved determining the transformation temperature of red lead by measuring the shift of the Raman peaks of β-PbO, which was mixed with red lead at different laser power densities.

The transformation laser power density required for red lead to transform into β-PbO at different temperatures was determined, and the corresponding Raman spectra are shown in Figure 5a–g. At room temperature, the transformation power density of the laser is 5.30 × 10^4^ W/cm^2^, (Figure 5a). However, when the temperature of red lead is increased to 50 °C, the transformation power density decreases to 3.51 × 10^4^ W/cm^2^ (Figure 5b). The transformation power density of red lead was 2.23 × 10^4^ W/cm^2^, 1.72 × 10^4^ W/cm^2^, 0.61 × 10^4^ W/cm^2^, 0.44 × 10^4^ W/cm^2^, and 0.40 × 10^4^ W/cm^2^ mW at temperatures of 100 °C, 150 °C, 200 °C, 250 °C, and 300 °C, respectively (Figure 5c–g). The transformation laser power density decreases with increasing temperature, indicating that the laser provides the additional energy required for the transformation of red lead at different temperatures. Figure 5h illustrates the fitted curve of the transformation laser power density required at different temperatures. When the laser power density is 0 W/cm^2^, the calculated thermodynamic temperature of red lead is approximately 831 K, as determined by the fitted curve Ln(*T*) = 0.95 × exp(−*p*/1.05) + 5.69, where *T* is the temperature and *p* is the laser power density, the unit for 0.95 and 5.697 is Ln(K), and the unit for 1.05 is 10^4^ W/cm^2^. This temperature is about 558 °C, which is 63 °C higher than the transformation temperature measured previously, indicating that the degradation of red lead under laser irradiation can be ascribed to the laser’s thermal effect.

However, the fitted curve is not good. The main influence is due to the error induced by the measurement. The laser power density can be affected by the focusing conditions, which can in turn affect the accuracy of the transformation power density required. Additionally, it is difficult to determine the appearance of β-PbO when reducing the laser power density. It seems that the transformation laser power density of red lead decreases rapidly as the temperature increases.

For further confirmation, the Raman peak shifts of β-PbO were used to estimate the temperature at the focus of the laser beam at various laser power densities. β-PbO, which is the degradation product of red lead, may affect the accuracy of temperature measurement. There are three primary reasons for the selection of β-PbO. Firstly, its Raman signals are very strong from 25 °C to 650 °C. Secondly, β-PbO is very stable from 25 °C to 650 °C. Thirdly, β-PbO and red lead have different Raman bands, making it convenient to observe the Raman shift under different temperatures. The Raman spectra of β-PbO at different temperatures were obtained (532 nm). Figure 6a presents the Raman mapping of β-PbO from 50 to 650 °C. All the Raman bands shift down to low wavenumbers and become weak with increasing temperature. The positions of the strong band at 289 cm^−1^ at different temperatures are shown in Figure 6b. Obviously, the relation between the Raman shift and 1/T is approximately linear. Hence, this relationship can be reflected by the following equation: ω = ω_0_ + αT, where ω_0_ = 289.0 cm^−1^ and α = −0.03445 cm^−1^/°C. Therefore, the temperature at the focus of the laser beam can be calculated using the Raman peak shifts of β-PbO.

It is known that the Stokes and anti-Stokes intensity ratios in the spectrum can directly provide the temperature information [13,14,15]. However, when we applied uniform spectral processing in our calculations, we noticed that the resulting temperature variation law and accuracy were poor. This issue was also observed in the laser-heated samples. The relevant temperature and calculation data are presented in Appendix A. The error between the actual temperature and the calculated temperature is small, below 210 °C. However, beyond 310 °C, the error increases significantly, and some of the calculated results are implausible. The position of the Raman peak is closely associated with temperature, and by excluding other factors such as pressure and applied voltage, we can establish the relationship between the peak positions and temperatures. In this study, since there are no other influencing factors apart from heating, the Raman peak shift can be used to calibrate and predict the temperature in the mixed sample.

Figure 7 shows the Raman spectra of the mixture of β-PbO and red lead at different laser power densities (532 nm). The precise peak position was obtained using the average value of the positions of anti-Stokes Raman lines and Stokes Raman lines. The band of β-PbO (289 cm^−1^, low laser power density) shifted down to low wavenumbers when the laser power density was increased. When the laser power density reached 5.1 × 10^4^ W/cm^2^, the band shifted to about 273.2 cm^−1^. With the help of the previous equation, the in situ temperature can be calculated to be about 459 °C. Therefore, the in situ temperature is very high when the laser power density reaches 5.1 × 10^4^ W/cm^2^. It is attributed to the heating effect of the laser that causes the transformation of red lead. However, this temperature is around 50 °C lower than the transformation temperature (495 °C). This difference is due to the different absorbances and thermal conductivities of red lead and β-PbO. As shown in Figure 8, the absorption spectra of red lead and β-PbO are different. The absorbance of red lead is very weak from ~800 nm to 600 nm and begins to become strong rapidly from ~600 nm to ~540 nm and is very strong from ~540 nm to 400 nm. However, the absorbance of β-PbO is also very weak from ~800 nm to 640 nm and becomes strong slowly from ~640 nm to ~430 nm and remains from ~430 nm to 400 nm. At the wavelength of 532 nm, the absorbance of red lead is ~0.5942 which is stronger than the absorbance of β-PbO (0.2008). The thermal diffusivities of red lead and β-PbO are presented in Table 2. The thermal diffusivities of red lead and β-PbO are 0.039 mm^2^/s and 0.068 mm^2^/s, respectively. Red lead has a lower thermal diffusivity than β-PbO, meaning it is easier to cause the accumulation of heat when irradiated by the 532 nm laser. Therefore, the temperature obtained from the Raman peak shifts of β-PbO mixed with red lead should be lower than the actual temperature of red lead in the mixture.

## 4. Discussion

Previous studies on the degradation of red lead have not thoroughly investigated the degradation mechanism or determined the appropriate laser power density [7,8]. Our study discovered that the transformation threshold of red lead under laser irradiation was 5.1 × 10^4^ W/cm^2^, and no significant changes were observed after irradiation at 2.3 × 10^4^ W/cm^2^ for 7200 s. At a laser power density of 5.1 × 10^4^/cm^2^, the temperature at the focal point reached around 450 °C, which is very close to the temperature at which red lead begins to degrade. The difference in temperature is caused by the different optical and thermal properties of the mixed samples. Due to its strong absorption and poor thermal conductivity, red lead can rapidly increase in temperature at the focal point, reaching approximately 500 °C under such low laser power density. Therefore, the degradation of red lead induced by laser irradiation can be attributed to the photothermal effect.

However, the obtained Raman spectra also carry temperature information corresponding to the measurement laser power, mainly reflected in the peak shift of the Raman bands. Other pigments such as azurite, malachite, and vermilion degrade when a 532 nm laser is used as the excitation source [17,18,19]. This type of pigment degradation is also attributed to the photothermal effect. Therefore, accurate Raman spectra could be obtained by adopting the proper laser power.

However, photochemical effects are also found in some pigments. Realgar, as a yellow pigment, degrades into orpiment when irradiated by a 532 nm laser [20]. Once realgar is irradiated by light, the degradation process begins, and it becomes faster when the light power is increased. In this case, we have to adopt an appropriate laser, such as a 785 nm laser, as the excitation source to avoid any damage to cultural relics while obtaining accurate spectra.

This information can be helpful in developing conservation and restoration strategies for artworks or artifacts containing red lead, where degradation is a concern. Additionally, the findings on the transformation power density of lead red at different temperatures may be useful for understanding and predicting red lead degradation under certain conditions. The absorption spectra of pigments may suggest a suitable laser for Raman measurement.

## 5. Conclusions

We conducted a study that revealed that even a low laser power can cause the degradation of red lead and lead to the formation of new substances. Through the use of micro-Raman spectroscopy, we investigated the degradation phenomenon of red lead induced by lasers. The degradation product was found to be consistent with the thermal degradation product, β-PbO. Moreover, we found that the threshold of degradation of red lead is approximately 5.1 × 10^4^ W/cm^2^. Our findings can be helpful in guiding measurement protection and avoiding any negative influences of degradation products on the accuracy of results. Our research also helped explain the mechanism behind the degradation of red lead caused by lasers. We demonstrated that it is primarily due to the photothermal effect, which can be explained by considering the strong absorption and very low thermal conductivity coefficients. This explanation fills a gap in our understanding of this phenomenon.

However, the thermal process of red lead is rather complicated and includes issues related to the intermediate product α-PbO, the structural morphological change of the sample, and the re-oxidation process of β-PbO. In further work, we will adopt in situ XRD, SEM, and micro-Raman spectroscopy to investigate the thermal decomposition and re-oxidation process of red lead in detail.

## Figures and Tables

**Figure 1 materials-17-00770-f001:**
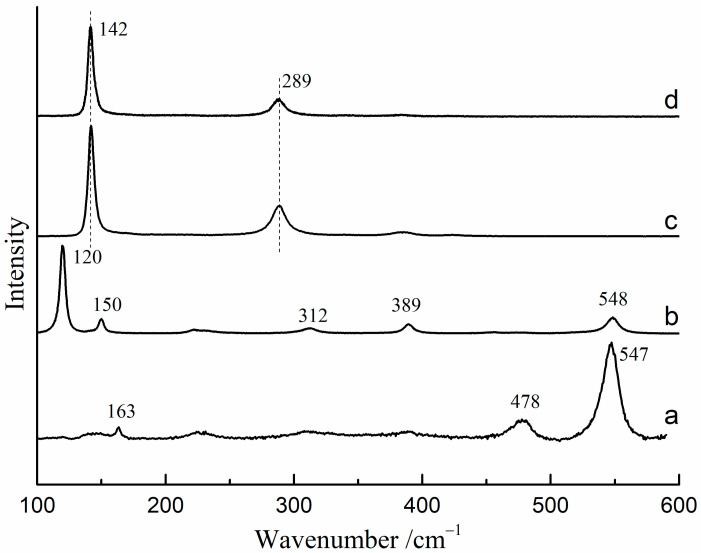
The Raman spectra of red lead and β-PbO under different excitation sources: (**a**) red lead under 532 nm excitation (2.3 × 10^4^ W/cm^2^), (**b**) red lead under 785 nm excitation, (**c**) β-PbO under 532 nm excitation, (**d**) β-PbO under 785 nm excitation.

**Figure 2 materials-17-00770-f002:**
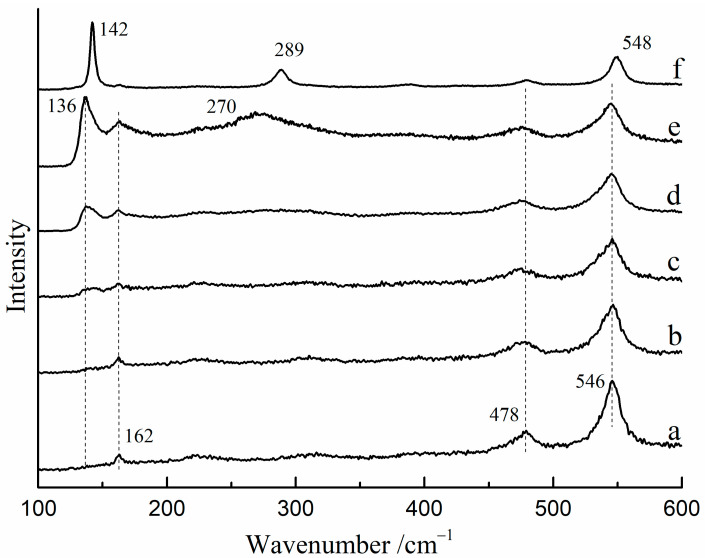
The Raman spectra of red lead after exposure to various laser power densities. (532 nm), (**a**) 3.2 × 10^4^ W/cm^2^, (**b**) 3.6 × 10^4^ W/cm^−2^, (**c**) 4.4 × 10^4^ W/cm^2^, (**d**) 5.1 × 10^4^ W/cm^2^, (**e**) 5.4 × 10^4^ W/cm^2^, (**f**) 0.2 × 10^4^ W/cm^2^ after irradiating with 5.4 × 10^4^ W/cm^2^.

**Figure 3 materials-17-00770-f003:**
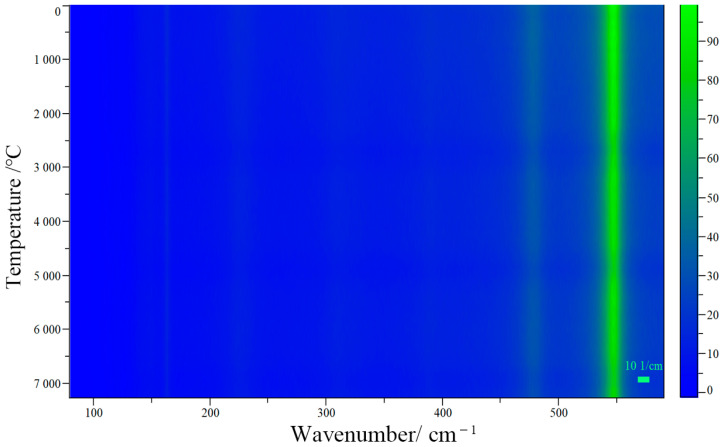
The Raman mapping of red lead under 2.3 × 10^4^ W/cm^2^ laser power density for a duration of 7200 s.

**Figure 4 materials-17-00770-f004:**
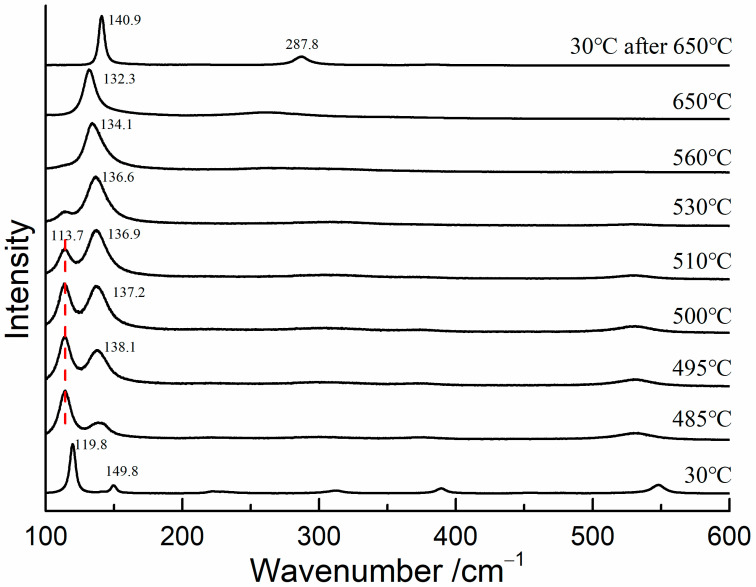
The Raman spectra of red lead at different temperatures (785 nm).

**Figure 5 materials-17-00770-f005:**
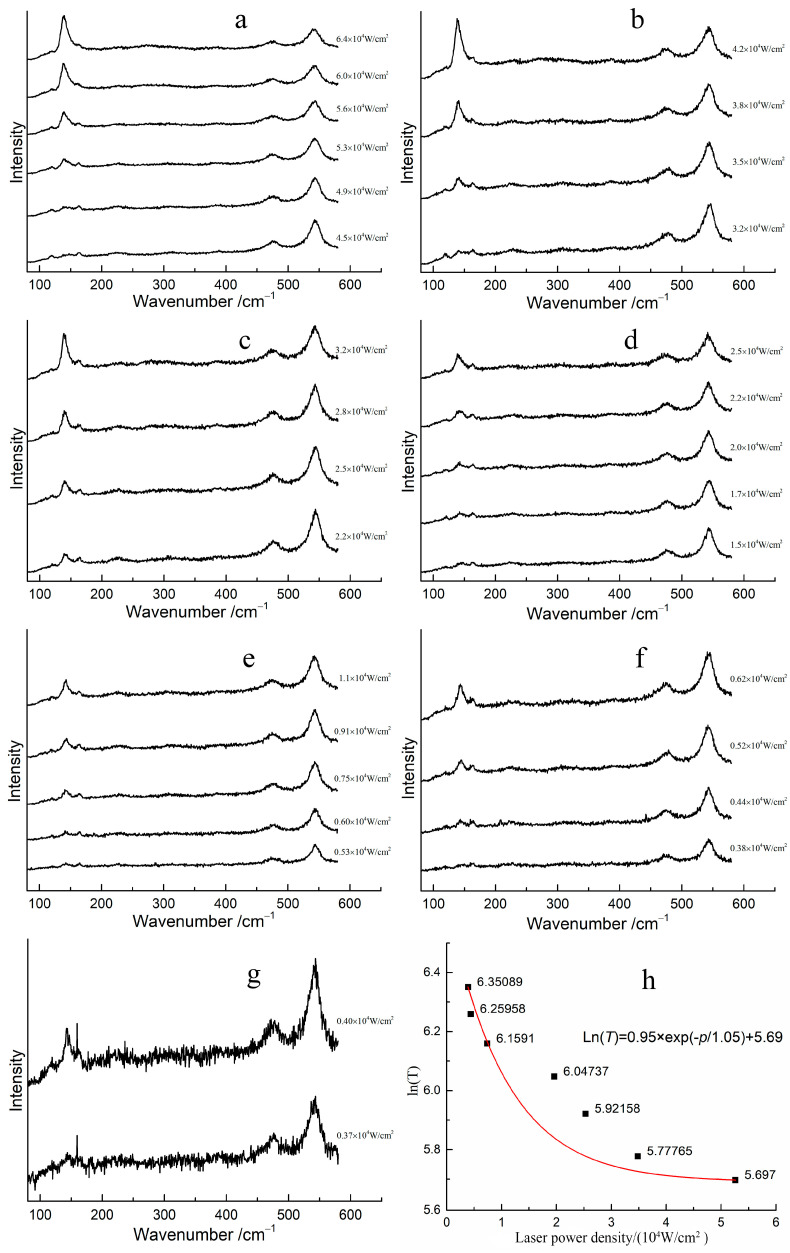
The Raman spectra of red lead under different laser power densities at various temperatures stages: (**a**) 30 °C, (**b**) 50 °C, (**c**) 100 °C, (**d**) 150 °C, (**e**) 200 °C, (**f**) 250 °C, (**g**) 300 °C, and (**h**) the relationship between the laser power density and Ln(*T*) (here, *T* is in Kelvin) at the earliest degradation of red lead.

**Figure 6 materials-17-00770-f006:**
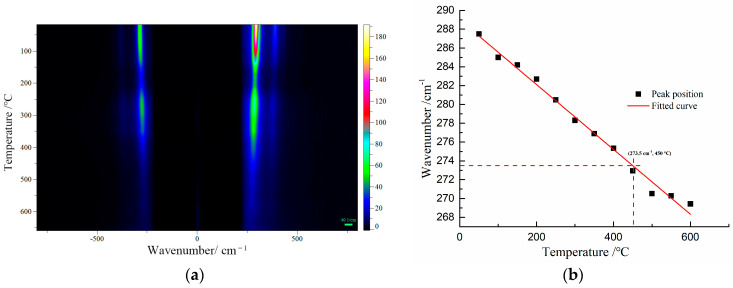
(**a**) The Raman spectra of β-PbO at different temperatures (532 nm excitation), (**b**) the Raman bands at 289 cm^−1^ of β-PbO observed at various temperatures.

**Figure 7 materials-17-00770-f007:**
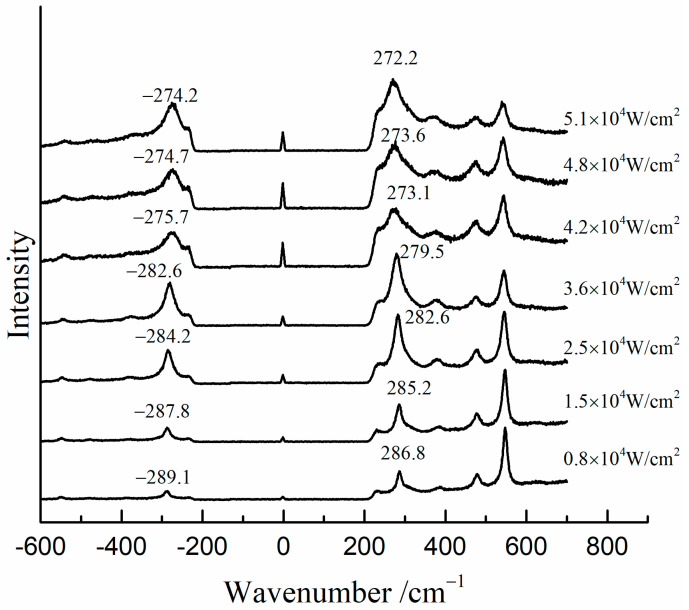
The Raman spectra of the mixture (β-PbO and red lead) at various laser power densities (532 nm).

**Figure 8 materials-17-00770-f008:**
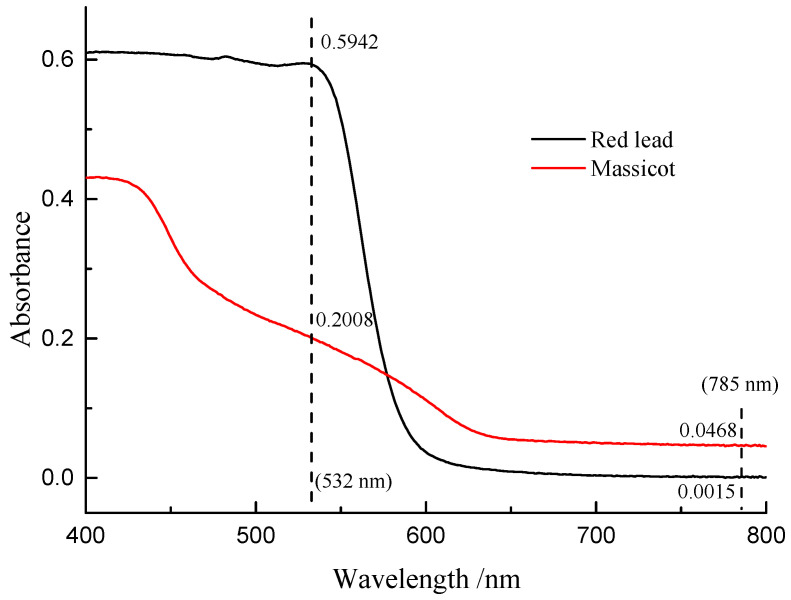
The absorption spectra of red lead and β-PbO.

**Table 1 materials-17-00770-t001:** The related parameters of Raman measurement during the experiment.

	Step 1	Step 2	Step 3	Step 4	Step 5	Step 6	Step 7
Material	Red lead, β-PbO	Red lead	Red lead	Red lead	Red lead	β-PbO	Mixture (Red lead, β-PbO)
Temperature/°C	22	22	22	30~650	30~300	30~600	22
Laser power density/10^4^ W/cm^2^	2.3 (532 nm)	3.2~5.4	3.2~4.2		0.37~6.4	0.53~1.1	0.34~0.62
Excitation wavelength	532 nm 785 nm	532 nm	532 nm	785 nm	532 nm	532 nm	532 nm

**Table 2 materials-17-00770-t002:** The thermal conductivity of red lead and β-PbO at room temperature.

Samples	Thermal Diffusivity (mm^2^/s)	Density (g/cm^3^)	Specific Heat Capacity (J/(kg·K)) [16]	Thermal Conductivity (W/(m·K))
Red lead	0.039	9.30	210	0.076
β-PbO	0.068	8.74	200	0.118

## Data Availability

Data are contained within the article.

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
