# Peer review of "Raman Study of 532-Nanometer Laser-Induced Degradation of Red Lead"

_materials, 2024, doi:10.3390/ma17040770_

Round 1

Reviewer 1 Report (Previous Reviewer 2)

Comments and Suggestions for Authors

I checked the revised version of your paper and I found that my comments and suggestions have been satisfied.

Author Response

Thanks for your review.

Reviewer 2 Report (New Reviewer)

Comments and Suggestions for Authors

The paper aims at evaluating the laser induced degradation of red lead under the Raman analysis.

The paper needs a deep review before publication. Here the major suggestions:

1-     The authors do not show a proper knowledge of Cultural Heritage field. Line 27: “…in order to protect them better.” Non-destructive methods in this field are used for the characterization of materials and of their conservation state, the knowledge of artist’s technique, the identification of the best conservation plan. Please, rephrase this sentence.

2-     Line 32: …“it exists as PbO2….” What’s the meaning of this sentence? Red lead exists as plattnerite???

3-     The entire introduction is confused. The authors should clearly state the aim of the study at the end of the introduction; for instance, they should also mention the use of 785nm excitation wavelength or the experiments with heating plate.

4-     A table with the design of the experiments is needed with indication of: pigment and parameters used (different laser power density, time, temperature, excitation wavelength….).

5-     Line 105 “…lower than the degradation energy…” which excitation wavelength? Line 108 “…temperature of red lead…” which excitation wavelength? Line 111 “…under different laser power density…”  which excitation wavelengths?

6-     Line 111 mixture of massicot and red lead, please specify the relative percentage of the two pigments in the mixture.

7-     Figure 2:  bands at 136 cm-1 and 270 cm-1 should be justified. Are they intermediate phases? Or what?

8-     Lines 157-160 please rephrase the explanation of photothermal process and photochemical process. It is too complicated for readers who are not familiar with this issue.

9-     Line 166 “A 785 nm laser was used as the excitation source to eliminate the influence of the 532 nm laser” . What’s the meaning of this sentence? It is not clear at all.

10- Lines 176-177: again, not clear sentence.

In general, it is not clear the real advantage of this study when the conclusions are that in the analysis of Cultural Heritage materials the laser power of Raman measurements should be kept low. This is well established in this field.

Comments on the Quality of English Language

Extensive editing of English language required

Author Response

Reviewer 3 Report (New Reviewer)

Comments and Suggestions for Authors

The work submitted for evaluation entitled: Raman study of 532 nm Laser-induced degradation of red lead concerns the important problem of degradation of the red lead trioxide pigment, commonly called red lead. It is a trade name eagerly used by various professional groups and has also entered scientific publications. The degradation of the red lead pigment Pb3O4 has been the subject of archaeometry research for years using various techniques to analyze materials. According to Aze et al. there are still many unexplained aspects of the degradation of this pigment.

The authors deal with the thermal degradation of Pb3O4, although they do not take into account numerous previous studies on the stability of this compound at elevated temperatures (450-500°C), which show that Pb3O4 (Pb2[PbO4] lead orthoplumbate) is subject to thermal decomposition in a single stage: Pb3O4 ® PbO.

It is generally known that lead compounds are unstable at elevated temperatures and therefore the following statement does not add anything new: Therefore, using low laser power density when measuring red lead in cultural relics 289 with a 532 nm laser can avoid damage and obtain accurate information about the pigment 290.

However, the experimental determination of the Pb3O4 degradation threshold under laser excitation should be considered an important achievement.

The authors show that the degradation of red lead pigment can be comprehensively analyzed using Raman analysis. However, this is a one-sided approach, and this work could be the first stage of research.

A serious shortcoming of the work is that these engineering studies are based on one research method - Raman analysis. It seems that supplementing these studies with results obtained from other research methods, such as DSC, EDX, SEM, XRD, would make this work more scientific and would allow for a better interpretation of the numerous research results presented. There was no thermodynamic analysis of the thermal decomposition reaction of Pb3O4, and only one work in this area was cited [Handy], and yet the thermal decomposition is influenced by both temperature and the partial pressure of oxygen. In addition, four known lead oxides have different colors: PbO yellow, Pb2O3 orange, Pb3O4 red, PbO2 brown - which can be used in optical research.

You can also point out to the authors that they use the mineralogical manner in their nomenclature. Instead of writing that the chemical compound Pb3O4 undergoes thermal decomposition to PbO, the authors write that "red lead" decomposes to massicot, although massicot is a mineral that is not obtained in this reaction. However, this approach to chemical problems is quite common not only in archaeometry. For this reason, this manner and the language of specialists must be respected.

In conclusion, I would like to emphasize that the work may be accepted for publication provided that it is treated as the first stage of research. The authors should declare that in further work they will make efforts and in the next stages they will enrich their work with optical, thermal and analytical tests (EDX, XRD).

Author Response

Reviewer 4 Report (New Reviewer)

Comments and Suggestions for Authors

Manuscript Materials-2801669

Raman study of 532nm Laser-induced degradation of red lead

Yan Li , Junjie Ma , Kang He , and Fengping Wang

deals with the iportant issue of the relation of Raman laser intensity and wavelength in analyses of pieces of art containing red lead.

The topic is up to date and the the method could be very intersting, buty the structure of the manuscript and the narration is extremely naive and it is omitting obvious control methods. The manuscript needs to be completely rethought. The issues are very redundanttly descirbed, a more clean style would help the Authors to find the logical flaws.

In particular

The chemical structure of the red lead and PbO2 should be depicted together with by products. This should be stated in the beginning of the manus together with the absorption spectra to motivate the choices of laser lines, and not vice-versa.

There is no trace of reflectance spectra or evaluation.

The samples need a thickness characterization for example by SEM, porosity, reflectance, purity. And different conformation of sample should be compared.

Massicot is natural PbO2 mineral, here it is inappropriate to reconstruct a mineral form,  as no X ray is produced one can say it is likely PbO2.

The temperature is not directly measured if not with the antistokes / stokes method (not shown as spectra). The method of choice is instead ossimoric as you consider the decomposition of in situ red lead as a temperature method used to determine the temperature of decomposition of red lead in situ.

The time of the laser irradiance is not well described: it seems that it is as long as having stable results. This should be assessed, and it should at least be considered that pulsed laser is as well possible in Raman spectroscopy.

Some minor typos:

Abstract

absorbance at approximately 0.5942 at 532 nm

absorbance of approximately 0.5942 at 532 nm

---

Line 25 : abundant pieces of art (“art” is the making, not the object)

Line 120 The stokes Raman

Line 121 using the 532

Line 124 using the 785

145-148 the caption is confusing

Comments on the Quality of English Language

The English is mostly correct if not for with some missing determinate articles, but the redundancy in the issues is dispersing the good part of the work.

Round 2

Reviewer 2 Report (New Reviewer)

Comments and Suggestions for Authors

Point 1: The authors do not show a proper knowledge of Cultural Heritage field. Line 27: “…in order to protect them better.” Non-destructive methods in this field are used for the characterization of materials and of their conservation state, the knowledge of artist’s technique, the identification of the best conservation plan. Please, rephrase this sentence.

Response 1: Thank for your valuable advice. We have rephrased this sentence. Non-destructive methods such as Raman spectroscopy, reflectance spectroscopy, and X-ray fluorescence spectroscopy are widely used in the field of cultural heritage for material characterization, assessing conservation state, understanding artist's techniques, and determining the most suitable conservation plan. However, the use of lasers in these methods may potentially result in material damage. Therefore, it is crucial to assess and prevent laser-induced damage to cultural relics.

Reply 1: why the authors say “the use of lasers in these methods” when in these methods XRF is included? Which laser is used in XRF technique? Please, be accurate with this point and rephrase again.

Point 3: The entire introduction is confused. The authors should clearly state the aim of the study at the end of the introduction; for instance, they should also mention the use of 785nm excitation wavelength or the experiments with heating plate.

Response 3: Thank for your advice. We have rewrite the aim of this study. In this study, we investigated the degradation threshold of red lead under different temperatures and laser power densities using microRaman spectroscopy at a wavelength of 532nm. We determined the in-situ temperature at which red lead starts to degrade, which was found to be consistent with the results obtained from variable temperature Raman spectroscopy using 785nm excitation. We further explored the causes of red lead degradation by measuring the absorption spectrum and thermal conductivity. These additional measurements helped us understand the underlying mechanisms behind the laser-induced degradation of red lead. This finding can provide valuable guidance for measurement protection and help avoid influence of degradation products on accurate results.

Reply 3: “…which was found to be consistent with the results obtained from variable temperature Raman spectroscopy using 785nm excitation”. This a result of your study. Please rephrase again splitting the aim of the study and the results achieved. This is still confused.

Point 4: A table with the design of the experiments is needed with indication of: pigment and parameters used (different laser power density, time, temperature, excitation wavelength….).

Response 4: Thank for your advice.We have added the laser degradation experiment in the revised manuscript.

Reply 4: I didn’t mean that. I meant a table with the indication of the parameters used in your experiments (what you described in the design paragraph). You can also keep the new laser degradation experiment description but it should be included in the experiment paragraph. So, re-done this point. ( Moreover, I didn’t find the citation of the table 1 in the text.)

Point 8: Lines 157-160 please rephrase the explanation of photothermal process and photochemical process. It is too complicated for readers who are not familiar with this issue.

Response 8: Thanks for your nice advice. We have rephrased the explanation of photothermal process and photochemical process in the revised manuscript.

The photochemical reaction that involves either the absorption or emission of radiation. The absorption of an photon often provides the energy required to break chemical bonds and initiate a reaction sequence. Generally the initial photochemical products are unstable and undergo secondary thermal and/or photochemical reactions, yielding more stable photoproduct. In photochemistry, reactions initiated from ground electronic states are thermal reactions, because their rates depend on temperature, as distinct from the photochemical reaction rates. Photothermal reactions lead to product generation only when thermodynamic and kinetic requirements are satisfied. Here, the discovery of the degradation threshold appears to meet the thermodynamic and kinetic requirements.

Reply 8: the sentence “The photochemical reaction that involves either the absorption or emission of radiation “ is unclear in English. In general, the entire paragraph has to be revised for English.

Point 11: In general, it is not clear the real advantage of this study when the conclusions are that in the analysis of Cultural Heritage materials the laser power of Raman measurements should be kept low. This is well established in this field.

Responses 11: Thanks your advice. Indeed, when conducting measurements on cultural relics, it is common knowledge to use a low laser power to minimize any potential damage. In this study, we that even such a low laser power can induce degradation of red lead and lead to formation of new substances. This finding can provide valuable guidance for measurement protection and help avoid influence of degradation products on accurate results. Additionally, the mechanism behind degradation of red lead caused by laser is not well understood. Through our research, we have demonstrated that it is primarily due to the photothermal effect, which can be explained by considering the absorption and thermal conductivity coefficients. This explanation fills a gap in our understanding of this phenomenon.

Reply 11: the authors should clearly explain that in the conclusion paragraph. It’s unclear in the actual version of the paper.

Comments on the Quality of English Language

Poor english

Round 3

Reviewer 2 Report (New Reviewer)

Comments and Suggestions for Authors

Although it is still unclear the impact of the paper to me, at least in this form can be published

Comments on the Quality of English Language

See above

This manuscript is a resubmission of an earlier submission. The following is a list of the peer review reports and author responses from that submission.

Round 1

Reviewer 1 Report

Comments and Suggestions for Authors

The paper documents an interesting area of the research which worth reading and gives a guideline for practical application. I left some comments to improve the paper. Read and work on the paper based on the comments.

1.      Please add the contribution of the paper to the abstract. It is not clear what is the novelty of this work.

2.      Please explain your design of the experiment.

3.      Table 2 “Thermal conductivity …” Please check the values.

4.      What are the governing factors provided in the discussion? This is important to bold this section for a better understanding of the readers.

5.      What solution you are offering using this experimentation?

6.      The conclusion needs to improve on the detail of the findings. Please reword and add more detail to the conclusion.

7.     The application of laser scanning is numerous these days and this device has some advantages in comparison with other tools. Authors are encouraged to mention about this note to clarify the contribution of their paper. Suggest reading and adding the following document. This enhances your paper and illustrates that this work has industrial applications. “Laser subtractive and laser powder bed fusion of metals: review of process and production features”.

Comments on the Quality of English Language

English needs some work.

Reviewer 2 Report

Comments and Suggestions for Authors

The paper “Raman study of 532 nm Laser-induced degradation of read lead” is an interesting communication that, after some improvement, can be considered for publication in the journal Materials.

I found the paper interesting and useful for the knowledge of pigment behaviour and for the degradation patterns, but I suggest to better focus the paper.

In the abstract and in the introduction the authors must explain the novelty of their work in respect to the literature and why their research is important in the field of cultural heritage (conservation, restoration knowledge of historical materials and techniques, etc.).

In the section 2.1, lines 68-74, please remove the part: “The Materials and Methods should be described with sufficient details to allow others to replicate and build on the published results. Please note that the publication of your manuscript implicates that you must make all materials, data, computer code, and protocols associated with the publication available to readers. Please disclose at the submission stage any restrictions on the availability of materials or information. New methods and protocols should be described in detail while well-established methods can be briefly described and appropriately cited”.

This comes from the original template of the journal.

The same in the section 3. Results from line 94 to line 96.

At page 3, lines 104-106 the sentence “The Raman bands of red lead are located at 120, 150, 312, 389, and 548 cm-1 when adopting 785 nm laser as the excitation, while the Raman bands are located at 163, 478, and 547 cm-1 under the excitation of 532 nm laser” need revision (are located … are located).

I suggest to move the supplementary materials in the main document, being this very short.

Discussion and conclusion must be increased. More comparison with similar studies in the literature must be reported and, in the conclusion the authors should write about the impact of their research in the field of materials science and cultural heritage.

Comments on the Quality of English Language

English language is clear.

I suggest only a moderate check.

Author Response

Please  see the attatchment.

Reviewer 3 Report

Comments and Suggestions for Authors

The manuscript covers an interesting topic, but seems to be submitted in a non-finished version. Also, data analysis lacks quality sufficient for publication – in particular Figure 3 and the corresponding text. For more reasons that follow below, I recommend rejection and resubmission.

Figure 2: The term on the ordinate should read “Intensity” and not “intersity”.

Figure 3 and line 155: The fitting curve is not well explained:

i)                    Why is the temperature given in °C and not in Kelvin?

ii)                   The denominator in the exponent should have a unit, shouldn’t it?

iii)                 Where does the value “24” indicate and shouldn’t it have a unit, too?

iv)                 There are no ticks/scaling on the ordinate (temperature). Please improve.

v)                   What is the origin to use an exponential function?

vi)                 The fit is not really good. Please show error bars and discuss thoroughly the discrepancy between measured data and the fit.

vii)               Can you also plot it on a logarithmic scale?

viii)              The word on the ordinate should read “absorbance” and not “abordance”.

Results, line 94 – 96:

This part is unbelievable. The authors simply forgot to delete the template text. Has this manuscript been proof-read before submission?

The second Figure 5 (absorption spectra (not spectrum) of Pb3O4 and PbO) should be termed Figure 6. Please change accordingly. What are the sample thicknesses? What about a spectrum of red lead and massicot instead of Pb3O4 and PbO?

In general, it is impossible to follow the temperature determination. Please provide more information.

The Discussion section of 12 lines (201 to 213) is somewhat ridiculous and unacceptable. Please write a thorough discussion section.

Line 223: Titles are missing for the figures.  

Supporting information: There is no title and no author list. Please add.

Several typos, e.g. intensity in line 84.

Comments on the Quality of English Language

see text above
